# Therapeutic radiographers' delivery of health behaviour change advice to those living with and beyond cancer: a qualitative study

Nickola D Pallin ,[1,2] Rebecca J Beeken,[1,3] Kathy Pritchard-Jones,[4] Laura Charlesworth,[5] Nick Woznitza,[6] Abigail Fisher[1]

¹Behavioural Science and Health, University College London, London, UK
²Department of Allied Health Sciences, London South Bank University, London, UK
³Leeds Institute of Health Sciences, University of Leeds, Leeds, UK
⁴London Cancer, London, UK
⁵University of Lincoln, Lincoln, UK
⁶Radiology, Homerton University Hospital NHS Foundation Trust, London, UK

**Correspondence to**
Nickola D Pallin;
pallinn@lsbu.ac.uk

## ABSTRACT

**Objectives** Therapeutic radiographers (TRs) are well placed to deliver health behaviour change advice to those living with and beyond cancer (LWBC). However, there is limited research on the opinions of TRs around delivering such advice to those LWBC. This study aimed to explore TRs' practices and facilitators in delivering advice on physical activity, healthy eating, alcohol intake, smoking and weight management.

**Setting and participants** Fifteen UK-based TRs took part in a telephone interview using a semi-structured interview guide. Data was analysed using the framework analysis method.

**Results** Emergent themes highlighted that TRs are mainly aware of the benefits of healthy behaviours in managing radiotherapy treatment related side effects, with advice provision lowest for healthy eating and physical activity. Participants identified themselves as well placed to deliver advice on improving behaviours to those LWBC, however reported a lack of knowledge as a limiting factor to doing so. The TRs reported training and knowledge as key facilitators to the delivery of advice, with a preference for online training.

**Conclusions** There is a need for education resources, clear referral pathways and in particular training for TRs on delivering physical activity and healthy eating advice to those LWBC.

## Strengths and limitations of this study

► This study provides an insight in therapeutic radiographers' views on all key modifiable health behaviours for those living with and beyond cancer.

► The participants worked in different radiotherapy departments, offering insight into the practices among therapeutic radiographers in the delivery of healthy behaviour advice from a wide range of hospitals.

► Whilst data saturation was reached, the sample size was small and therefore the findings may not be representative of the views of the wider therapeutic radiography workforce.

► The response rate was low (20.8%), therefore the participants might be more interested in the role of health behaviours in cancer survivorship, which might bias the responses towards a positive view on the role of therapeutic radiographers in delivering advice within their role.

and psychosocial outcomes after a cancer diagnosis.[2–8]

Despite the potential benefits of healthy behaviours, few people LWBC are meeting the WCRF recommended health behaviour recommendations.[9 10] Those LWBC report one key reason for not adopting healthier lifestyle behaviours is lack of advice and support from their healthcare team.[11] Healthcare professionals (HCPs) are well placed to bring about positive health behaviour changes among cancer patients.[12] A trial of brief advice among breast cancer survivors showed that a simple physical activity recommendation from a HCP doubled the percentage meeting national exercise guidelines.[12] Despite this, research to date among both HCPs and those LWBC consistently shows that few oncology HCPs offer guidance to oncology patients on healthy lifestyle behaviours.[13–21]

Reported barriers among HCPs in providing health behaviour advice for those

## INTRODUCTION

It is estimated that 40% of cancer cases are linked to unhealthy behaviours.[1] Based on evidence from systematic literature reviews and meta analyses, the World Cancer Research Fund (WCRF) recommend that individuals are physically active, limit consumption of energy dense foods, salty foods, red meat and avoid processed meat, eat more plant foods, maintain a healthy weight, limit alcoholic drinks and avoid tobacco to reduce their risk of cancer.[2] Those living with and beyond cancer (LWBC) are also advised to follow these guidelines, due to increasing evidence that healthy behaviours may improve physical

LWBC, include believing that giving advice was not part of their role, lack of time with patients, lack of referral programmes, lack of resources such as education leaflets for those LWBC and lack of knowledge regarding guidelines and research findings.[16–21] A recent qualitative study with 21 oncology HCPs identified that advice on health behaviours provided to those LWBC focussed on general health and controlling side effects, with few HCPs advising on health behaviours in the context of improving survival outcomes.[20] While these studies provide useful insight into the practices and barriers among oncology HCPs the participants within these studies were primarily oncologists and nurses and focussed on the provision of physical activity and weight management advice. There is limited research on the opinions of therapeutic radiographers (TRs) in delivering advice on health behaviours to those LWBC, despite at least 50% of cancer patients receiving radiotherapy as part of their cancer treatment.[22]

TRs are the only health professionals qualified to deliver radiotherapy and play a central role in supporting cancer patients.[23] In the UK, the College of Radiographers recognise the importance of TRs in providing health behaviour advice to improve patient outcomes.[24] TRs are also seen as an integral part of the health force in driving improvements in well-being as outlined in the 2017 publication of "AHPs into Action, using Allied Health Professionals to transform health, care and well-being" which states that radiographers are key to implementing a preventative healthcare approach and that their expertise should be used to design and deliver health interventions.[25] TRs are ideally placed to deliver health behaviour advice, particularly through Making Every Contact Count (MECC).[26 27] MECC is a strategy whereby health professionals use every appropriate opportunity and interaction with patients to promote healthy behaviours and signpost to relevant healthcare services using an 'Ask, Advise, Act' framework.[27] MECC fits extremely well within the TRs' role in which patient education is a key part of radiotherapy practice with TRs providing care to the same patient every day, often over a number of weeks.[23] TRs therefore have the potential to make significant contributions in supporting positive health behaviour changes among those LWBC.

However, despite these opportunities one survey in the UK among 102 TRs identified that TRs rarely advise patients on the key modifiable health behaviours including smoking, alcohol, healthy eating and exercise.[15] The findings also showed lack of knowledge and training as barriers among TRs in delivering advice on these topics.[15] Similarly, focus group interviews with 38 TRs identified that lack of knowledge and training were barriers to the provision of smoking cessation advice.[28]

Challenges remain in translating behaviour change interventions into existing care pathways and practices in a way that is appropriate for use by health professionals.[29] Understanding TRs' practices, and what support they need in delivering advice on the topics of physical activity, healthy eating, alcohol intake, smoking and weight management could inform the development of interventions that will enable TRs' in delivering advice on improving health behaviours to those LWBC. Qualitative research is appropriate for exploring the beliefs, experiences and motivations of individuals on specific matters and allow for more information and clarification.[30] Limited qualitative data exists on TRs' practices and views on delivering advice on these health behaviour topics. This study therefore aimed to address this and through a qualitative methodology explore TRs' practices in delivering health behaviour advice, in addition to exploring the facilitators in delivering such advice. Preferences regarding training on delivering this advice were also explored.

## METHODS

### Participants and recruitment

Participants were TRs working in the UK in a clinical role. They had provided their contact details on a previous online survey investigating TR's practices in delivering health behaviour advice agreeing to be invited for a follow-up telephone interview. An email was sent with an information sheet explaining the research and inviting these TRs. Those who agreed to take part signed a consent form prior to the telephone interview.

### Data collection

Semi-structured individual telephone interviews were carried out between April and May 2019 by a lecturer in therapeutic radiography with an MSc who had completed qualitative interviewing as part of their training (NP). The interviewer had no previous relationship with the study participants. The topic guide (see online supplementary material 1) was based on the guide used within a previous study[17] which explored oncology HCPs views on the provision of lifestyle advice to cancer patients. This guide was adapted for use among TRs, with additional questions added to assess preferences for training on delivering advice. The topic guide was piloted with two participants to check for comprehension of the questions. This data was included in the analysis because no substantial changes were required. The interviews lasted approximately 30 min (range: 20 to 40 min) and were audio-recorded, anonymised and transcribed verbatim. The transcripts were verified by NP against each recording to confirm accuracy. The aim was to carry out interviews until data saturation was reached. It was anticipated that 10 participants would be required to reach data saturation because it was a homogeneous group.[31] After 10 interviews were carried out, they were transcribed verbatim. Following familiarisation with the data NP generated the initial codes and it appeared that saturation was reached after 10 interviews as no new codes occurred in the 10th interview.[32] A further five interviews were carried out to confirm this.

**Table 1** Participant identifier and demographic characteristics

| Participant identifier | Demographic characteristics | |
| --- | --- | --- |
| | Gender | Professional grade |
| TR 1 | Female | Band 6 |
| TR 2 | Female | Band 7 |
| TR 3 | Female | Band 7 |
| TR 4 | Female | Band 7 |
| TR 5 | Female | Band 7 |
| TR 6 | Female | Band 7 |
| TR 7 | Female | Band 7 |
| TR 8 | Male | Band 8 |
| TR 9 | Female | Band 7 |
| TR 10 | Female | Band 7 |
| TR 11 | Female | Band 6 |
| TR 12 | Female | Band 7 |
| TR 13 | Male | Band 5 |
| TR 14 | Female | Band 7 |
| TR 15 | Male | Band 8 |

TR, therapeutic radiographer.

### Patient and public involvement

Patient input was not used in the design of the research methods. However, the topic guide was piloted with TRs in the academic setting. Additionally, the topic guide was piloted with two participants and these were included in the analysis.

### Analysis

The interview transcripts were analysed using the framework analysis method.[33] This method was chosen because it is an appropriate method for analysing homogeneous data and semi-structured interview transcripts, it is also appropriate when using inductive qualitative analysis.[33] A random selection of transcripts (n=3) were independently coded by AF to check for reliability. The researchers NP, AF and RJB met and agreed on a final coding list in an iterative process (AF and RJB are both experienced qualitative researchers and health psychologists). These agreed set of codes formed the analytical framework which was then applied to all of the transcripts and the data summarised in a matrix using Microsoft Excel. Themes were generated by reviewing the matrix connecting the data between the participants and the codes. The completed consolidated criteria for reporting qualitative research checklist is available in the online supplementary material 2.[34] Themes are presented in the results with supporting quotes and the participants identifier (table 1).

### RESULTS
### Participants

The radiotherapy radiography workforce census in the UK only reports the workforce's professional grade, and no other demographics.[35] In the UK, TRs' level of professional skills and knowledge are categorised by agenda for change professional band grades 5 to 8.[36] Therefore, in this study, the participants gender and professional grade were collected; no other demographic information was collected (table 1). The response rate to taking part in the interview was 20.8%. Seventy-two TRs were emailed and invited to take part in an interview, 15 returned consent forms and completed the telephone interview. Fifteen interviews were conducted with 12 women and 3 men. The participants came from all regions of the UK, including England, Wales, Scotland and Northern Ireland.

### Themes

Five main themes were identified: (1) TRs provide behaviour change advice to manage radiotherapy-related side effects; (2) TRs make judgements about when it is appropriate to deliver health behaviour advice; (3) Knowledge and training are key facilitators in the delivery of health behaviour advice; (4) TRs feel patients undergoing radiotherapy treatment seek guidance on health behaviours; and (5) TRs identify themselves as well placed to give health behaviour advice to patients.

### TRs provide behaviour change advice to manage radiotherapy-related side effects

Most respondents reported that they only provided advice on health behaviours that they believed would minimise radiotherapy-related side effects. This meant smoking cessation and alcohol intake were the two health behaviours TRs mainly advised on.

> With head and neck patients, we give advice, particularly on smoking and drinking, obviously get worse side effects (TR 6).

> The only thing we do generally say is about drinking plenty of fluids, avoiding alcohol. But that's more to do with prostate side effects, bladder reactions and reducing gas (TR 14).

> Radiographers are comfortable talking about alcohol when it comes to managing side effects (TR 12).

No TRs reported advising patients on healthy eating. Some TRs mentioned advising patients on dietary intake but this is to patients who are at risk of losing weight, for side effect management and potential impact on accuracy of radiotherapy treatment delivery.

> Healthy eating, I don't tend to discuss too much. A lot of patients have difficulty eating and we are encouraging maintaining weight while on treatment (TR 5).

> I'm not very sure if healthy eating is important. Any patients where we're treating, lower GI or pelvis, we would advise them to avoid very high-fibre foods, spicy foods, that might make them have very loose bowels, but other than that we say more or less keep on your same diet. We wouldn't generally discuss a healthy diet as a standard for all patients. No (TR 15).

Some TRs mentioned they advise patients to be physically active, however this was only in the context of managing radiotherapy and cancer-related fatigue.

> So exercise is one of my main ones that I focus on with all patients, particularly to help with their fatigue (TR 9).

> Exercise, I say that's its quite beneficial to help with fatigue (TR 12).

> I guess when we have patients come in, fatigue is one of the side effects, so we encourage our patients to remain active (TR 15).

### TRs make judgements about when it is appropriate to deliver health behaviour advice

TRs explained only discussing health behaviours, particularly smoking and alcohol with patients if there were evident indications of a problem. TRs also often reported making a judgement of whether appropriate to advise a patient on a particular health behaviour.

> So, quite often you can tell if a patient is a smoker, you can smell it, or you can tell by their skin (TR 11).

> I tend to give advice when you make a judgement of when it's appropriate, an example might be if a patient smelt of smoke (TR 12).

> Had patients come in and will smell of alcohol and at that time I'll say to the patient that it can exacerbate side effects (TR 15).

This meant TRs did not provide advice on health behaviours to every patient.

> But for those patients where it's not clearly going to benefit them to stop drinking, you would just mention it very briefly. Not every patient will have that information (TR 5).

### Knowledge and training are keys facilitator in the delivery of health behaviour advice
#### Delivery of advice matched by knowledge
The reported delivery of advice on health behaviours appeared to be matched by knowledge of the benefits among those LWBC.

One participant explained how he only appreciated the importance of physical activity in cancer survivorship after attending a talk and being made aware of the evidence.

> My experience of appreciating the role exercise was from attending a talk. I suppose it was really just highlighting in the studies the benefits obviously of a healthy lifestyle and introducing physical activity for patients on treatment (TR 15).

Healthy eating was a topic the participants felt particularly unqualified to deliver advice on, and reported lack of knowledge as a barrier to the delivery of advice on healthy eating.

> It's a difficult one, diet, I think. It's more a knowledge thing. If you don't have the knowledge about what you can and can't say, you're just not going to approach the subject (TR 12).

### A need for continuous postgraduate online training
All interviewed said they would welcome postgraduate training on delivering health behaviour advice. The majority expressed a preference for online training to help overcome the barrier of limited time among TRs to attend training.

> Online, you're not having to take time out of clinical practice, online is more accessible (TR 6)

Participants also mentioned that online training allows for yearly updates and continuous professional development.

> I think it'd be good (online training) because you can do it in your own time. Because I think that's sometimes the problem. You have this training once and then maybe it never gets brought up again. So it would be quite handy to have something small, every year, alongside all your other mandatory training (TR 12).

Participants did acknowledge that face-to-face training allows for further questioning that's not possible with online training.

> I think one-to-one training, because you can ask questions that may not be covered within the online training (TR 1).

To overcome the barrier of not all staff being able attend face-to-face training participants suggested it would be useful to train some TRs through face-to-face methods that they could then cascade to other TRs within the radiotherapy department.

> Maybe some face-to-face with some staff, that they could cascade down, might be useful as well (TR 4).

### A need for training in the undergraduate setting
It was also suggested to incorporate training on delivering lifestyle advice into the undergraduate education programme.

> Certainly, get it into the undergraduate course to start with, making them aware it is part of the role (TR 9).

> It's still not something that I can say was primarily covered in the undergraduates' training about the benefits of healthy lifestyle, you know, there's no real formal education that I can see (TR 15).

### TRs reported knowledge of resources and referral pathways as facilitators in the delivery of advice
Participants also felt knowledge of how to refer patients onto further support would enable them to have conversations on improving health behaviours. With some

TRs reporting that lack of knowledge of resources and referral pathways are barriers to initiating a conversation on behaviour change.

> There needs to be more information available to professionals of where exactly you can refer patients to, whether that be website, whether that be an app (TR 13).

> That's the only reason why they [therapeutic radiographers] don't want to open these conversations up, because they don't know where to go with it or how to refer on (TR 9).

They also acknowledge that in the short time they have with patients if they had a resource then would be more inclined to advise.

> Having something on a piece of paper, education and having the resources, if you can do it in 2 min you should be able to slip that in (TR 2).

> You don't always have that information at hand, so if it was readily available, I think we'd give out a bit more [health behaviour information] if it was just the case of pointing them in the right direction that would be a quick and easy thing to do (TR 3).

### The benefit of incorporating patients' perspective into training

Participants also mentioned that getting patients' perspectives on receiving advice on improving health behaviours should be incorporated into training.

> I think that would be better coming from the patients themselves, rather than just feedback from what journals, and other literature says (TR 7).

> If there'd even be patients that would be willing to maybe just even be involved with staff training (TR 3).

### TRs feel patients undergoing radiotherapy treatment seek guidance on health behaviours

Many of the TRs also described that patients often ask them for guidance around health behaviour changes, particularly on diet and exercise. This shows that patients see TRs as credible sources of information on health behaviours.

> We are getting asked the question more and more about weight loss, healthy living, wanting to exercise more (TR 4).

> It is quite a common thing to be asked at the end of treatment, not so much the smoking and alcohol, I have to say, but diet and exercise is certainly something that people commonly ask (TR 3).

### TRs identify themselves as well placed to give health behaviour advice to patients

TRs acknowledged that they are a consistent healthcare member for patients undergoing radiotherapy and have many opportunities to deliver lifestyle advice. Therefore,

TRs recognised that they are well placed to deliver health behaviour advice to patients.

> We're in a unique position because we do see the same patient day after day and you do kind of start to develop a relationship with them (TR 10).

> I think we're well placed to help influence patients' behaviours and it's something we should be seen to encourage and report (TR 7).

> We're in the best position where we see the patients, for a number of weeks every day to encourage any changes (TR 8).

From the interviews it appeared that many patients undergoing radiotherapy, excluding those at risk of malnutrition or significant weight loss, are primarily reviewed and assessed by TRs. This highlights that TRs are in an ideal position to deliver advice on health behaviours, particularly when asked about nutrition advice delivery.

> They routinely see the specialist radiographer, for the breast patients. But they don't have a dietitian appointment (TR 12).

> Prostate and breast are two tumour groups that are fully radiographer-led review and about 70% to 80% of our work load. They generally wouldn't be sent to a dietician (TR 15).

> Only have a dietitian on board for the head and necks (TR 9).

## DISCUSSION

TRs in this study saw themselves as well placed to deliver health behaviour advice, but also reported that they do not routinely provide advice to all patients. TRs were particularly unlikely to provide advice on healthy eating and physical activity, and were more likely to provide advice on those behaviours they believed would minimise radiotherapy or cancer-related side effects. This is in line with previous research among TRs.[15 28 37] In one qualitative study a key facilitator reported among TRs in delivering smoking cessation support to patients was knowledge of the link between smoking and toxicity.[28] Another qualitative study that explored allied health professionals' views regarding the provision of dietary advice to patients, highlighted that TRs report giving dietary advice to help counteract the side effects of radiotherapy.[37] Additionally, in our study, if TRs did provide dietary advice, this tended to be general advice rather than cancer-specific advice on healthy eating.

In some studies, oncology HCPs have reported they do not self-identify as the right person to provide lifestyle advice.[17 20] However, in this study TRs identified themselves as being well placed to deliver health behaviour advice and in a unique position as a consistent member of the multidisciplinary team providing care to patients. However, despite this, they do not feel qualified to deliver

advice, particularly on the topic of healthy eating. In the UK, poor diet has the biggest impact on the National Health Service budget, greater than alcohol consumption, smoking and physical inactivity.[38] It has been noted that there are insufficient dietitians to provide dietary advice to all patients who may need dietary support.[39] In response to this all HCPs are being asked to implement a preventative healthcare approach within their role and the delivery of healthy eating advice is fundamental to this.[23 24 40 41] Key to achieving this is that TRs will have the skills, knowledge and behaviours to improve the health and well-being of individuals.[24] As with other oncology HCP groups[16 17 20 37] this study identifies the need for education and training among TRs in delivering health behaviour advice, particularly on healthy eating and physical activity. This training should also address when and how to refer to other support if necessary, as this was identified as a key facilitator in the delivery of advice on health behaviours, particularly when time is a barrier to the delivery of this advice.[15–17]

All interviews demonstrated that TRs would welcome training on delivering health behaviour advice and recommended it as a key facilitator in delivering advice, in addition to incorporating it into the undergraduate setting. The need for postgraduate training among TRs in delivering health advice has also recently been reported by Charlesworth *et al*[28] in relation to the delivery of smoking cessation advice. Our findings from this study provide additional insight into TRs preferences on the type of training on delivering lifestyle advice to those LWBC, with TRs demonstrating a preference of online training in the postgraduate setting. Among HCPs online education has been reported to be as effective as face-to-face education.[42] Additionally, the use of online learning enables HCPs to carry out training at time that fits in with clinical work.[43 44] TRs in this study identified this benefit of online learning in overcoming the limited time available for TRs to undertake continuous professional development and additional training. Interestingly, TRs in this study mentioned having patient input in the training would be helpful. While HCPs input is key to the development of interventions, patient members play key advocacy roles and their input can enhance the outcomes of interventions.[45] Patient input may also help overcome the reported barrier of fear of causing offence to a patient, which has been reported as a barrier among oncology HCPs in delivery of health behaviour advice.[17]

Those LWBC wish to receive advice on health behaviours from their healthcare team[13 20 46] and is of particular importance as the period following a cancer diagnosis has been shown be a teachable moment and an ideal opportunity to motivate patients around the importance of healthy eating and physical activity.[47 48] This was made apparent in this study, whereby some TRs mentioned that healthy eating and exercise were the health behaviours patients ask for advice on more often, generally towards the end of their treatment. This further highlights the importance of supporting TRs in delivering evidence-based health behaviour advice to meet patients' needs.

TRs have a responsibility to educate patients on the importance of following healthy behaviours given the increasing evidence showing implementing healthy behaviours improve a number of physical and psychosocial outcomes after a cancer diagnosis.[2 3] Among premenopausal and post-menopausal women living with and beyond breast cancer, a systematic literature review and meta-analysis of 82 follow-up studies (n=213 075 breast cancer survivors) identified that being overweight increases the risk of all cause and breast cancer mortality.[4] Being physically active after a cancer diagnosis is also correlated with improved survival and reduced recurrence.[5 6 49] While data is limited, emerging research suggests healthy dietary behaviours after a diagnosis may improve outcomes.[3 50] In a prospective observational study of 1009 patients with stage III colon cancer, a higher intake of a typical Western diet was associated with a three-fold increased risk of disease recurrence and a 2.3-fold increased risk of all-cause mortality.[8] Additionally, those LWBC are at increased risk for developing cardiovascular disease, osteoporosis and diabetes and healthy behaviours can reduce the risk of developing these diseases.[51 52]

Of those interviewed in this study it appeared that those with breast, prostate and colorectal cancer are primarily reviewed and assessed by TRs. Therefore, it is the responsibility of TRs to deliver advice on improving health behaviours to these patients. This is also particularly important because the strongest evidence for the benefits of diet and exercise is currently in breast, prostate and colorectal cancer survivors.[53] These are also the most common cancers in the UK and radiotherapy plays a key role in managing these cancers.[22 54] Therefore, with the right skills and knowledge, TRs could deliver advice on improving health behaviours. By supporting self-efficacy among patients towards the end of their treatment which very often is in the radiotherapy department can be empowering for patients. Among those with prostate cancer, implementing dietary changes brought psychological benefit, as a method of coping and regaining control over their diagnosis.[46]

### Strengths and limitations

This is the first qualitative study among TRs to explore the provision of advice on all key modifiable lifestyle behaviours for those LWBC as per recommendations.[2] While the aim of qualitative research is not to generalise the findings the sample size was small, and therefore the findings may not be representative of the views of the wider therapeutic radiography workforce. However, data saturation was reached, likely due the homogeneous sample of participants. Additionally, the participants worked in different radiotherapy departments and therefore provide insight into the practices among TRs in the delivery of healthy behaviour advice from a wide range of hospitals. Also, the participants worked in cancer centres in England, Wales, Scotland and Northern

Ireland, providing insight into the practices across the UK. Another limitation of this study is the low response rate (20.8%) and that the participants might be more interested in the role of health behaviours in cancer survivorship, which might bias the responses towards a positive view on this topic and the role of TRs in delivering advice within their role. Despite this however, provision of health behaviour change advice was low, suggesting TRs may be even less likely to educate patients around the importance of healthy behaviours.

## Future research

This study highlights the need for training and education among TRs on the delivery of health behaviour advice to cancer patients, both in the undergraduate and postgraduate setting. Particularly on the topics of physical activity, healthy eating and weight management. Higher Education Institutions have a responsibility in educating the Allied Health Professional workforce on implementing health promotion within their role.[55] Further research among pre-registration TR students and lecturers within therapeutic radiography should therefore explore how best to address this need. Future research among TRs should also use purposive sampling to identify the views and health promotion practices among those who may not have a primary interest in the area of health behaviours among those LWBC.

## CONCLUSION

In conclusion, while the majority of TRs delivered some advice on health behaviours as part of their role, advice was mainly on smoking and alcohol intake. Most believed in the value of this advice in managing radiotherapy and cancer-related side effects. Provision of advice was lowest for weight management, healthy eating and physical activity. The findings show a need for training among TRs in delivering advice on improving health behaviours among those LWBC, with TRs reporting a preference for online training in the postgraduate setting.

**Acknowledgements** The researchers are grateful to the health professionals who participated in the study.

**Contributors** NDP had the original idea for the study and obtained the funding with AF, RJB and KPJ. NDP developed the design of the study, acquired the data, analysed and interpreted the data, drafted and revised the article and approved the final manuscript submitted. AF provided behavioural science expertise, contributed to the development of the study design and the recruitment approach, analysed and interpreted the data, reviewed, edited and approved the final manuscript. RJB provided behavioural science expertise, contributed to the development of the study design and the recruitment approach, interpreted the data and reviewed, edited and approved the final manuscript. KPJ provided oncology expertise and intellectual input into the recruitment approach, design and approved the final manuscript. LC provided therapeutic radiography and public health expertise which informed the development of the study design and reviewed the article and approved the final manuscript. NW provided radiography and public health expertise and contributed to the development of the study design, reviewed the manuscript for important intellectual content and approved the final manuscript submitted.

**Funding** This research was funded by Cancer Research UK (ref no: C59769/A25965).

**Competing interests** None declared.

**Patient and public involvement** Patients and/or the public were not involved in the design, or conduct, or reporting, or dissemination plans of this research.

**Patient consent for publication** Not required.

**Ethics approval** The methodology for this study was approved by the Human Research Ethics committee of University College London (reference 12945/001).

**Provenance and peer review** Not commissioned; externally peer reviewed.

**Data availability statement** All data relevant to the study are included in the article or uploaded as supplementary information. Data can be obtained by the corresponding author on reasonable request.

**Author note** This work was undertaken in the Department of Behavioural Science and Health, University College London, London, UK. The lead author was based there as a pre-doctoral research fellow.

**ORCID iD**
Nickola D Pallin http://orcid.org/0000-0002-1707-1353

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
