## [Reviewer comments · BMJ Open]

ARTICLE DETAILS

TITLE (PROVISIONAL)	Therapeutic radiographers' delivery of health behaviour change advice to those living with and beyond cancer: a qualitative study.
AUTHORS	Pallin, Nickola; Beeken, Rebecca; Pritchard-Jones, Kathy; Charlesworth, Laura; Woznitza, Nicholas; Fisher, Abigail

VERSION 1 - REVIEW

REVIEWER	Paul Shepherd Ulster University United Kingdom
REVIEW RETURNED	07-May-2020

GENERAL COMMENTS	The paper explores the novel realm of health behaviour change advice (Health promotion and education) delivered by therapeutic radiographers during the radiotherapy treatment pathway. The important professional role for therapeutic radiographers is discussed and the requirement for learning and training identified. The wording in the title is a little cumbersome and tends to detract from the immediacy of understanding although in retrospect the title meaning is understood. The reason for the study is clearly set out and the aim justified. A succinct explanation and justification of the methodology is offered and the target population and sampling technique explained. Data recording measures are established to endeavour to reduce bias and enhance the scientific rigour and trustworthiness of the findings. The paper acknowledges that the therapeutic radiographer motivation to participate in the study may create an element of bias within the responses. That said, the findings, even if from therapeutic radiographers with an inherent interest in patient support and wellbeing, clearly identify gaps and obstacles impacting on the professional effectiveness of therapeutic radiographers fulfilling the role. This opens the debate with respect to what further pre- and post- registration learning and training might be needed, implies that more effective multi-disciplinary teams in radiotherapy departments should be established in order to facilitate seamless referral when required and suggests a wider professional evaluation of health and lifestyle interventions will be needed in order to fully empower therapeutic radiographers to engage in health promotion and education for the benefit of the cancer patients. Overall a novel, well-designed and executed study, well-structured and organized, providing a catalyst for enhancement of professional practice and role development for therapeutic radiographers.
---

REVIEWER	Diana Greenfield Sheffield Teaching Hospitals NHS FT, UK
REVIEW RETURNED	18-May-2020

GENERAL COMMENTS	The authors aim to explore the current role of the Therapeutic Radiographer in delivering health behaviour change and what further measures are required to support them to achieve this. The authors found Therapeutic Radiographers' focus on health behaviour change related to immediate relevance to their patients, that is smoking and alcohol behaviour changes which are known to impact on acute radiotherapy side effects. Therapeutic Radiographer participants in the study reported they less frequently advised on other health behaviour change factors (such as weight management and physical activity). The paper is well written, clear and concise, and is a good and relevant literature in the context of current national policy. The authors have used qualitative research and have clearly explained the methodology and methods and this is appropriate. The paper makes a contribution to the literature since Therapeutic Radiographers are in a key position to advise cancer patients: half of whom receive radiotherapy. The rationale is provided in the context of the potential contribution Therapeutic Radiographers can make towards the "Making Every Contact Count's principle, where Therapeutic Radiographers may have an opportunity to influence and advise patients on a range of modifiable risk factors. The response rate to this survey was poor (15/72 (20.8%)) which means the findings may not be representative of the wider workforce. More so, the authors recognise, as a significant weakness, that the participants have self-selected with an interest in the subject. This major flaw in the study limits the robustness of the conclusions as it stands. The authors would have added strength to this study had they 1) done purposeful sampling of participants to include for example, current Therapeutic Radiography pre-registration students and Therapeutic Radiography lecturers and academics at Higher Education Institutions to explore further the principles and rational of Therapeutic Radiographer providing broader health behaviour change to cancer patients as a fundamental role component. Further the authors would have strengthened the study to put their main findings as, for example, a survey back to the original cohort, to ask them the relevance of the group and whether they concur with the findings. I recommend the authors acknowledge the poor response rate. The authors should also provide a rationale for why they didn't follow this course of action and, additionally, make recommendations regarding implications for further research. They should also strengthen the caution regarding drawing conclusions from this study given its major flaws. Finally, there is no sense of a further rationale for Therapeutic Radiographer providing health behaviour change for reasons other than to influence acute radiotherapy side effects. There is an increasing evidence base that cancer patients, including and sometimes specifically, radiotherapy patients are at increased longer term and late effects risks of both recurrence and of cardiovascular and other problems which may be influenced by modifiable health behaviours. The authors do not mention this at all and this would strengthen the rationale and should be included in the discussion. Whilst this study is limited by the significant weakness of its self-selected participants, it is the start and not the final word on the subject, and is therefore of sufficient interest for publication.
--

VERSION 1 – AUTHOR RESPONSE

Changes made in response to comments made by Reviewer 1

- Title amended
- In response to the comment "opens the debate to what further pre- and post- registration learning and training might be needed" the need for further research within the education setting has been added to the manuscript.

Changes made in response to comments made by Reviewer 2

- The low response has been added as a limitation to the manuscript and has been acknowledged within the limitations that the findings may not be representative of the wider workforce.
- In response to the potential of the participants self-selecting and having an interest in the subject, this has been acknowledged in the limitations.
- Recommendations for further research to use purposeful sampling has been added.
- Recommendations for further research among Therapeutic Radiography pre-registration students and Therapeutic Radiography lecturers and academics at Higher Education Institutions has been added to the manuscript.
- Further rationale and supporting evidence for Therapeutic Radiographer providing health behaviour change advice in the context of health behaviours improving outcomes and minimising risk of developing chronic health conditions among those living with and beyond cancer have been added to the discussion.

VERSION 2 – REVIEW

REVIEWER	Diana Greenfield Sheffield Teaching Hospital and NHS FT
REVIEW RETURNED	10-Jun-2020

GENERAL COMMENTS	The authors have made most of the changes to the manuscript as recommended in my first review. For future it would be helpful for the reviewer to have further details of the changes made and navigated to the changes in the manuscript. In terms of the lower response rate, this has not been adequately addressed in this revision and I recommend the actual wording and calculation of the response rate is provided in the results section (15/72 (20.8%)) and that this is also added to the strengths and limitations section on P4.
--

VERSION 2 – AUTHOR RESPONSE

1. The wording and calculation of the response rate has been added to the strengths and limitations section on page 4 (Lines 17 - 26)

"Whilst data saturation was reached, the sample size was small and therefore the findings may not be representative of the views of the wider therapeutic radiography workforce."

"The response rate was low (20.8%), therefore the participants might be more interested in the role of health behaviours in cancer survivorship, which might bias the responses towards a positive view on the role of therapeutic radiographers' in delivering advice within their role."

2. The wording and calculation of the response rate has been added to the results section (page 10, Lines 3 - 8)

"The response rate to taking part in the interview was 20.8%. Seventy-two TRs were emailed and invited to take part in an interview, 15 returned consent forms and completed the telephone interview."

3. The low response rate has been added to the limitations at the end of the manuscript (page 20, Lines 32 - 37 & 48 - 60).

"Whilst the aim of qualitative research is not to generalise the findings the sample size was small, and therefore the findings may not be representative of the views of the wider therapeutic radiography workforce."

"Another limitation of this study is the low response rate (20.8%) and that the participants might be more interested in the role of health behaviours in cancer survivorship, which might bias the responses towards a positive view on this topic and the role of TRs in delivering advice within their role."

4. Previous suggestions for revisions are highlighted in red on pages 19-21